

# Extension of the WRF-Chem volcanic emission pre-processor to integrate complex source terms and evaluation for different emission scenarios of the Grimsvötn 2011 eruption

Marcus Hirtl[1,2], Barbara Scherllin-Pirscher[1], Martin Stuefer[2], Delia Arnold[1,3], Rocio Baro[1], Christian Maurer[1], Marie D. Mulder[1]

[1]Zentralanstalt für Meteorologie und Geodynamik, Vienna, A-1190, Austria
[2]Geophysical Institute, University of Alaska Fairbanks, Fairbanks, AK 99775, USA
[3]Arnold Scientific Consulting, Manresa, 08242, Spain

*Correspondence to*: Marcus Hirtl (marcus.hirtl@zamg.ac.at)

**Abstract.** Volcanic eruptions may generate volcanic ash and sulfur dioxide ($SO_2$) plumes with strong temporal and vertical variations. When simulating these changing volcanic plumes and the afar dispersion of emissions, it is important to provide the best available information on the temporal and vertical emission distribution during the eruption. The volcanic emission module of the chemical transport model WRF-Chem has been extended to allow integrating detailed temporally and vertically resolved input data from volcanic eruptions. The new emission pre-processor is tested and evaluated for the eruption of the Grimsvötn volcano in Iceland 2011. The initial ash plumes of the Grimsvötn eruption differed significantly from the $SO_2$ plumes posing challenges to simulate plume dynamics within existing modelling environments: observations of the Grimsvötn plumes revealed strong vertical wind shear that led to different transport directions of the respective ash and $SO_2$ clouds. Three source terms, each of them based on different assumptions and observational data are applied in the model simulations. The emission scenarios range from (i) a simple approach, which assumes constant emission fluxes and a pre-defined vertical emission profile, to (ii) a more complex approach, which integrates temporarily varying observed plume top heights and estimated emissions based on them, to (iii) the most complex method that calculates temporal and vertical variability of the emission fluxes based on satellite observations and inversion techniques. Comparisons between model results and independent observations from satellites, lidar and surface air quality measurements reveal best performance of the most complex source term.

## 1 Introduction

In the past decades, there have been several eruptions with significant impact on aviation (e.g., Albersheim and Guffanti, 2009; Guffanti et al., 2010; Bolić and Sivčev, 2011). Airspace closure or flight re-routing has been required since volcanic ash may cause significant damage to turbine engines when internal fans are exposed to elevated concentration levels over certain time periods (Clarkson et al., 2016). During the eruption of the Eyjafjallajökull volcano in 2010, wide areas of the European airspace were closed for days (Bolić and Sivčev, 2012). From 15 until 22 April 2010, 104 000 flights were





cancelled (Alexander, 2013). In May 2011, the Grimsvötn eruption led to a cancellation of 1 % (~900 of total ~90000) of planned flights in Europe during a period of two days (http://www.volcano.si.edu/reports/). Observational data, e.g., from radar, lidar or satellite were used to observe locations and extent of volcanic clouds. Numerical model simulations were performed by Volcanic Ash Advisory Centers (VAACs) to predict the dispersion of the volcanic ash and $SO_2$ clouds in
support of emergency management. After the Eyjafjallajöküll 2010 eruption, harmonized thresholds were defined for aircraft alerting procedures and provided by the London and Toulouse VAACs to support the Volcanic Ash Contingency Plan (VACP, Edition 2.0.0 – July, 2016).   Low, medium, and high contamination regions were defined for volcanic ash mass concentrations: less than or equal to 2 mg/m³, greater than 2 mg/m³, less than or equal to 4 mg/m³, and higher than 4 mg/m³, respectively.
Characterizations of emission source terms during volcanic events are typically extremely challenging to obtain, and best model results can only be achieved by integrating all available observational data. Volcanic source terms include the source strength, its vertical and temporal variations as well as size, density and shape of emitted particles. A realistic estimate of the source term is crucial to accurately predict the transport of ash and gases released during volcanic eruptions.

The Weather Research and Forecasting (WRF, Grell et al., 2005) model coupled with Chemistry (WRF-Chem) is able to
realistically simulate the dispersion of ash clouds from volcanic eruptions (e.g., Webley et al., 2012; Stuefer et al., 2013; Hirtl et al., 2019). However, the standard volcanic emission pre-processor of WRF-Chem has some deficiencies degrading the model performance related to the dispersion of volcanic ash and $SO_2$ clouds. These deficiencies can be mainly attributed to limitations of the description of temporal and vertical variability of emission fluxes (Hirtl et al., 2019). In other words, the WRF-Chem volcanic emission application has been limited, to using source terms based on "simple" mass eruption rate
timeseries. This study presents the extension of the WRF-Chem volcanic emission pre-processor towards more complex source terms and evaluates the results for the eruption of the Grimsvötn volcano in Iceland in May 2011.

The Grimsvötn volcano is one of the most active and well-known volcanoes in Iceland (e.g. Gudmundsson and Björnsson, 1991, Vogfjörd et al., 2005, Witham et al. 2007, Moxnes et al., 2014). Over the past centuries, it has erupted about once per decade. During the most recent major eruption, which occurred from the 21 until 25 May 2011, significant amounts of $SO_2$
and ash were injected into the atmosphere. The Grimsvötn plume development was observed by GOME-2 (Global Ozone Monitoring Experiment-2, Flemming et al., 2013), OMI (Ozone Monitoring Instrument, Sigmarsson et al., 2013), IASI (Infrared Atmospheric Sounding Interferometer, Carboni al., 2013; Moxnes et al., 2014), SEVIRI (Spinning Enhanced Visible Infra-Red Imager, Cooke et al., 2014), AIRS (Atmospheric Infrared Sounder, Chahine et al., 2006), AATSR (Advanced Along-Track Scanning Radiometer) and MODIS (Moderate Resolution Imaging Spectroradiometer, Tesche et al.,
2012). This study used observations from the IASI, SEVIRI, AATSR and AIRS instruments. The IASI observations are used in the Bayersian inversion technique to calculate a volcanic ash and $SO_2$ source term and the SEVIRI, AIRS and AATSR for evaluation purposes. Beside satellite observations, for evaluation of the WRF-Chem model output are also used lidar and ground station measurements from national air quality monitoring networks.



Figure 1 shows ash and SO$_2$ clouds observed by the IASI instrument for the 23 May 2011. The comparison between ash and

SO$_2$ observations clearly reveals different dispersion patterns. While SO$_2$ was first transported to the north of Iceland and then towards Greenland and the Canadian and U.S. east coast, volcanic ash was transported to the south of Iceland and then towards northern U.K. and eastern Scandinavia. The separation of the ash and SO$_2$ clouds was caused by different injection heights and vertical wind shear (Moxnes et al, 2014). Forecast models, which did not take into account the different release heights at the early stage of the eruption, produced unrealistic forecasts as shown by comparisons to satellite data (Tesche et

al., 2012; Cooke et al., 2014). Prata et al. (2017) provided observational perspectives on the event and advised using separate source terms for ash and SO$_2$. Following this, we used several source terms for the Grimsvötn eruption to drive WRF-Chem and for validation of results.

For this study, three scenarios with each a different source term, based on different assumptions and observational data, are applied in the model simulations. The emission scenarios range from (i) a simple approach, which assumes constant emission

fluxes and a pre-defined vertical emission profile, to (ii) a more complex approach, which integrates temporarily varying observed plume heights and estimated emissions based on observed plume heights, to (iii) the most complex method that calculates temporal and vertical variability of the emission fluxes based on satellite observations and inversion techniques.

The remainder of this paper is divided into five sections: section 2 provides a technical description of the extension of the WRF-Chem volcanic ash emission pre-processor. Section 3 describes the WRF-Chem model setup and emission scenarios.

The results and model evaluation with different observations (satellite, lidar and surface air quality measurements) can be found in section 4. Summary and conclusions are given in Section 5.

## 2 Extension of the volcanic pre-processor of the WRF-Chem model

The WRF-Chem model simulates emission, transport, mixing, and chemical transformation of trace gases and aerosols simultaneously with the meteorology. The model enables to use various options for dynamic cores and physical

parameterizations (Skamarock et al., 2008). The online approach (meteorology with air chemistry) accounts for a numerically consistent air quality forecast; no interpolation in time or space is required.

In the official release of WRF-Chem v4.2 (code is available at https://github.com/wrf-model/WRF/releases), volcanic emission sources can be considered only in a very simplified way. The model can simulate the dispersion of volcanic emissions specified by the initial plume height, erupted mass (ash and SO$_2$), the duration of the eruption, and the aerosol bin

size distribution (up to 10 bin sizes).

If erupted mass is not known, it can be calculated applying the Mastin formula (Mastin et al., 2009), which relates plume height $h_{plume}$ (in meters) to the emitted mass per timestep $m_{emitted}$ (in kg/s)

$$m_{emitted} = 2600 * (0.0005 * h_{plume})^{4.1494} \qquad (1)$$






For the vertical source term structure, a 75/25 umbrella shaped plume is applied: 25 % percent of the mass from vent height to a certain height (~73 % of plume height) of the plume, then 75 % of the mass distributed to a parabolic distribution until plume top height. For real-time applications, this is a straightforward approach, as the development of the volcanic emission cannot be predicted.

Stuefer et al. (2013) had extended the volcanic emission pre-processor with time variant emissions, one could either specify mass fluxes directly or only apply the Mastin equation for different time steps based on the plume height (implemented in WRF-Chem version 3.4).

We extended the WRF-Chem capability towards user-defined volcanic source emission data that is read in through an external file and adjusted to the model grid. These emission specifications (in kg/m/s) should vertically resolved time series

of ash and $SO_2$, as shown in Table 1. The date and time entries refer to the start of the emission interval, and the specified height (above ground level, AGL) refers to the lower limit of the height interval. Emissions of the last time step and the top-most level are the upper bounds (therefore set to zero) of the highest sub-column and the last time step. The emission specifications could be produced by any suitable method, as long as they are available in the format provided here. They could for example be produced with Bayesian inversion techniques, as included in the third emission scenario in this study.

As the emission fluxes have to be provided at heights above ground, the pre-processor (linearly) interpolates the input values for each column to the model levels of WRF-Chem (see Fig. 2). Depending on the difference between the model terrain height of the vent and the real vent height, an offset can be defined to account for deviations due to the limited model resolution. Finally, the resulting total volcanic emission, which is used for the WRF-Chem simulation, is scaled; in order to compensate for the deviations caused by the interpolation to ensure mass conservation. The routines have already been used

in the frame of a volcanic eruption exercise for an artificial eruption of the Etna (Hirtl et al., 2020).

## 3 WRF-Chem model simulations

### 3.1 Model setup

WRF-Chem simulations were performed from the 21 to 26 May 2011. The model domain extended from northern Africa to the north of Greenland and from eastern Newfoundland to western Russia. Model resolution was 12 km horizontally and 47

levels vertically from the surface up to 50 hPa. Meteorological fields used as initial and boundary conditions were derived from the European Centre for Medium-range Weather Forecasts (ECMWF). Parameterization of physical processes included the Mellor-Yamada Nakanishi and Niino Level 2.5 PBL scheme (Hong et al., 2006), the Grell three-dimensional (3D) ensemble cumulus parameterization (Grell and Freitas, 2014), and the Rapid Radiative Transfer Model for Global (RRTMG) long-wave and short-wave radiation schemes (Iacono et al., 2008).

All simulations considered 10 volcanic ash bins and $SO_2$ (chem_opt = 402). Total fine ash was assumed to be composed of the finest 4 bins (the other bins were set to zero): 12.7 % of particles within 0.01 to 3.9 μm in diameter, 18.2 % within 3.9 to 7.8 μm, 29.1 % within 7.8 to 15.6 μm, and 40.0 % within 15.6 to 31.0 μm. This is consistent with the FLEXPART (FLEXible





PARTicle dispersion model; Stohl et al., 2005) model simulations that are used as input for the Bayesian inversion (emission scenario 3) to calculate an a posteriori source term (see section 3.2). It uses the size distribution which represents the bin-size

range to which the IASI satellite observations are mainly sensitive to.

### 3.2 Volcanic emission scenarios

Three emission scenarios (further designated as S1, S2, and S3) were selected to test the sensitivity of ash and $SO_2$ dispersion to volcanic emissions. Underlying complexity of the source terms ranges from a very simple first guess to a

sophisticated a posteriori source term, which was derived with satellite observations and inverse modeling.

Simple volcanic emission source terms can be derived from the eruption plume height (Mastin et al., 2009; see also Section 2). During the Grimsvötn eruption in 2011, plume height measurements were performed with weather radars (e.g., Petersen et al., 2012) and made available by the Icelandic Meteorological Office (IMO). The timeseries of observed plume heights AGL from the Keflavik radar is shown in Fig. 3.

The first emission scenario (S1) used only the first observed plume height (15 km) and assumed constant emissions of ash and $SO_2$ for the eruption which was assumed to last 2 days. This is a very rough estimate, though a common approach to get a first idea of the dispersion of the volcanic plume. The associated uncertainties increase rapidly, in particular if eruption characteristics change. An ash emission rate of about 0.01 Tg/s was estimated with the Mastin formula (Eq. 1) for ash. For $SO_2$, the total emitted mass was assumed to be 1 Tg, yielding a constant emission rate of about 5787 kg/s for the two days.

The vertical source term structure was modeled as a 75/25 umbrella shaped plume.

The second emission scenario (S2), was based on the entire observed plume height time series. The same plume heights were assumed for ash and $SO_2$ even though Prata et al. (2017) found that observed plume heights were more linked to $SO_2$ than to ash. Ash emission rates were computed with the Mastin equation in the first step for each time step. Based on the total amount of IASI ash and $SO_2$ measurements for the 4 days of the eruption, the hourly emission rates were further constrained

with these satellite observations following Moxnes et al. (2014). The total emitted mass used in the simulations was scaled to 0.4 Tg for ash and for $SO_2$ the time series of ash was constrained to 0.36 Tg $SO_2$. The magnitude of the $SO_2$ emission is reasonable, as shown by Flemming et al. (2013), who estimated a total emitted mass of $SO_2$ of 0.32 Tg. After scaling, volcanic emission rates ranged from 67 kg/s to 12080 kg/s for ash and from 60 kg/s to 10872 kg/s for $SO_2$. The vertical structure of the source term was again modeled as 75/25 umbrella shaped plume but considering different plume heights.

The third emission scenario (S3) uses the source terms derived of $SO_2$ and ash produced with the Bayesian inversion technique, using FLEXPART runs and observations from the IASI instrument. The source term files were provided by Moxnes et al. (2014), who also described the method in detail. The source terms are shown in Fig. 4, with a vertical resolution of 1000 m. In contrast to S1 and S2, the vertical structure of these emissions does not follow an umbrella-shaped plume. While maximum $SO_2$ emissions (up to 11541 kg/s) were found at altitudes between about 5 km and 12 km ASL in the

morning of the 22 May, ash emissions were largest (7539 kg/s) at lower altitudes (below approximately 2 km ASL) in the morning on 23 May.





According to Fig. 4, the highest ash emissions are below 5 km ASL, while the $SO_2$ emission peaks are located at attitudes between 5 km and 13 km. Figure 5 summarizes the emission rates of all three scenarios. Compared to S2 and S3, S1 for $SO_2$ seems to be an average estimate, in contrast to S1 for volcanic ash, which assumes much higher emissions than the other

scenarios (logarithmic scaled y axis). The highest ash emissions occur after the 22 May, 20:00 UTC, the $SO_2$ emissions have already decreased at this time.

**3.3 Model inter-comparison of predicted ash considering aviation regulation aspects**

To evaluate the performance of the three emission scenarios in a first step, the model runs are intercompared for the first two days of the eruption. Focus is set on ash concentration levels, which are important for aviation aspects. All regions with

volcanic ash mass concentration greater than or equal to 4 mg/m³ are considered "high contamination" areas (ICAO, 2016). Passenger aircraft are advised not to fly through regions of volcanic ash concentrations that exceed 4 mg/m³. This threshold is therefore most important for aviation aspects.

Figure 6 shows the 4 mg/m³ contour lines of maximum sub-column (between WRF-Chem model levels) volcanic ash for all emission scenarios for the 22 and the 23 May, 00:00 and 12:00 UTC. Since emission rates of scenario S1 are much higher

than those of S2 and S3 (see Fig. 5), ash-rich regions are distinctively larger for S1 than for the other scenarios. This is best visible on 23 May, 12:00 UTC, when the S1 cloud spreads from Greenland and Iceland towards the UK. Neither the S2 nor the S3 scenario show any significant area with an ash concentration exceeding 4 mg/m³. This illustrates how crucial it is to carefully estimate the emission rates. Comparison between scenarios S2 and S3 reveals a higher ash concentration on 22 May for S2, but lower ash contamination on the day after. This can be explained with corresponding emission rates (Fig. 5).

An evaluation of the source-term performance and investigation of corresponding ash and $SO_2$ dispersion from all WRF-Chem simulations can be found in the next section, where model runs will be compared with independent observations.

**4 Evaluation of WRF-Chem simulations with observations**

**4.1. Comparison of volcanic ash and SO₂ with satellite data**

In this section, the model simulations are compared to satellite observations of ash and SO2 from different instruments.

SEVIRI is an instrument onboard the geostationary METEOSAT satellite, which observes any point within its field of view every 15 minutes (over Europe every 5 minutes), AATSR was an instrument onboard of ENVISAT (mission ended in 2012), which was in a sun-synchronous orbit with an equator crossing time of 10:00 local time. Several studies exist in which data of the two instruments have been used to analyse volcanic eruptions. Virtanen et al. (2014) have developed a (plume and cloud) height estimate algorithm for AATSR, which has been validated and compared to other satellite-based instruments

and in situ data. The method was applied to the Eyjafjallajökull eruption 2010 and performed reasonably well. Kylling et al. (2015) compared SEVIRI with IASI observations for the Grimsvötn eruption and found deviations in mass loadings of about a factor of 2 between the instruments, with the higher concentrations measured by SEVIRI, for the plume going northward.





Results from scenario S1 are not considered here because the model-intercomparison already indicated a strong overestimation of ash simulated with S1 (Fig. 6). The model simulations for the scenarios S2 and S3 are compared to total

column ash from SEVIRI and AATSR observations for 23 May 2011 in Fig. 7. The ash cloud was observed south of Iceland by both SEVIRI and aerosol optical thickness (AOT) from the AATSR, which were in good agreement. The simulation based on S3 performs well and reproduces the location of the cloud. The maximum total ash concentration based on S3 was higher than that of SEVIRI with 10.7 $g/m^2$ compared to 3.9 $g/m^2$, respectively. Based on S2, however, highest ash concentrations (maximum 4.5 $g/m^2$) are simulated in the north-west of Iceland due to wrong assumptions of the emitted ash plume top

heights. On the next day (24 May, not shown) the scenarios S2 and S3 further drift apart, again with S3 being in better agreement with the observations. While most of the observed cloud moves towards the east (to the UK and Scandinavia), SEVIRI also detected some ash north of Iceland, which is assumed to be noise in the data, not present in AATSR and in the model. Ash mass loading of the cloud northeast of Scotland is as high as 3.7 $g/m^2$ in SEVIRI data and 0.8 $g/m^2$ in the S3 model run.

Observations from the AIRS instrument, a hyperspectral imager on the polar orbiting EOS/Aqua satellite, are used for comparison of $SO_2$. The AIRS has a spatial resolution of 13.5 km and has already been used to study other volcanic eruptions, such as the Etna eruption in 2002, published by Carn et al. (2005). They showed that comparisons with MODIS observations indicated that AIRS is likely to underestimate $SO_2$ in the vicinity of the volcano due to the presence of dense ash.

Simulated $SO_2$ concentrations from all WRF-Chem runs compared to the AIRS are shown in Fig. 8 for the 23 May 2011. Two days after the eruption the $SO_2$ cloud was transported further towards the north. All model scenarios reproduce this pattern in general but show differences in plume width and in the distance of the plume from the vent. AIRS data also showed $SO_2$, which was transported towards the east, this part could not be reproduced by the model simulations. The maximum observed $SO_2$ concentration was about 95 DU, which was detected northwest of Iceland. Highest $SO_2$

concentrations from model simulations range from about 60 DU in S2 to 910 DU in S1. During the next days (not shown) the highest values simulated by the model deviate more and also the location of the maximum value shows relevant differences.

The comparison of the WRF-Chem simulations with satellite observations revealed, that the proper prediction of the location of the ash plume for the Grimsvötn 2011 eruption is only possible for this test case when the source terms for ash and $SO_2$

are treated separately.

## 4.2 Comparison with ground-based observations

Measurements from two lidar stations and several ground-based in-situ observations are used to further evaluate the S3 model simulation. Both S1 and S2 based simulations, did not result in relevant ash concentrations at these locations.



### 4.2.1 Lidar profiles at selected stations

Vertical profiles of volcanic ash are compared with measurements from lidars (pink dots in Fig. 9) in Stockholm (Tesche et al., 2012; Tesche et al., 2007; Althausen et al., 2009) and Cabauw.  During 24 May, the model simulates that, a narrow, elongated band of ash was transported over northern European mainland. The cloud ranged from the Netherlands up to northern Scandinavia (Fig. 9). It slowly approached Stockholm (Fig. 9 northern pink dot), where maximum ash column concentrations were found at about 23:00 UTC.

The lidar measurements in Stockholm (Fig. 10) revealed ash arrival a few hours later, on 25 May between 03:00 and 04:00 UTC. The temporal offset is, however, relatively small, considering that Stockholm is far away from the source region and that the ash cloud has already been transported for a couple of days.

The vertical profile of modeled maximum ash concentration (average from 24 May 19:00 UTC until 25 May 03:00 UTC) based on S3 over Stockholm is below the ash mass concentration estimates from Tesche et al. (2012) that were based on lidar

measurements between 02:00 and 08:00 UTC. While the model predicts maximum ash concentrations (<100 µg/m³) within a thick vertical layer between 500 m and 2500 m, the lidar observations revealed a sharp peak at about 1000 m with values between 50 µg/m³ (lower estimate) and 450 µg/m³ (upper estimate). The maximum (as well as the minimum) curve is based on different assumptions when calculating the mass from the extinction coefficient. According to Tesche et al. (2012), the minimum curve is more likely to represent the real observed ash concentrations.

Volcanic ash was also detected by the lidar at Cabauw on 25 May, 16:30 UTC. Figure 11 shows a qualitative comparison of the observed backscattering coefficient profile and the S3-based modeled ash concentration profile, both normalized to 1. Both data sets clearly show enhanced aerosol concentrations between about 500 m and 2000 m with the peaks at 1250 m and 1500 m in the lidar and model data, respectively. The predicted vertical extension of the ash layer shows a very good agreement with the observation at the Cabauw station.

### 4.2.2 Comparison with PM10 observations at selected ground stations


For the days of the Grimsvötn eruption, surface measurements of PM10 are available from several stations in northern Europe (orange dots in Fig. 9). These data have already been used by others to investigate the eruption and to evaluate dispersion models (e.g., Prata and Prata 2012; Tesche et al. 2012; Moxnes et al. 2014). The WRF-Chem output (the finer 3 ash bins corresponding to the size range of PM10) from scenario S3 was interpolated to the station locations and compared

for the 2 days of the volcanic ash cloud overpass on the 24 and 25 May. Figure 12 shows the time series of the observed hourly data for the stations.

In general, the observed PM10 concentrations are slightly higher than the model prediction. This is not only true for PM10 peaks, when a large portion of PM10 can be attributed to volcanic ash, but also for the entire time period.  This model bias is caused by missing anthropogenic and biogenic aerosol emissions as well as secondary aerosol formation yielding zero PM



concentrations before and after the volcanic ash overpass. These contributions were not considered in the simulations as the emphasize of this study was on the ash and $SO_2$ emitted by the volcano.

For most of the ground stations, the plume arrival is simulated well by the model, although the model underestimates the observations. In Aberdeen, a temporal shift of about 6 hours is observed. At this station the modelled peak is later compared to the observed peak; in contrast to the station in Oslo where the simulated peak arrives about 6 hours earlier. The ground

observations in Stockholm reveal that the time of the plume arrival is captured by the model very well in contrast to the lidar observation, which indicates a temporal shift of about 4 hours.

It has to be acknowledged that to simulate the dispersion over long distances of elevated source emissions is a hard task for a model, especially when the concentrations are compared to "point measurements" at the ground far away from the source. Uncertainty in the emission and the meteorology (e.g. vertical mixing) have a strong impact on the dispersion and causes

deviations between model and observations especially for this complex case.

## 5. Conclusions

The developments presented in this paper permit integrating complex source temporally variable profiles into the emission pre-processor of the WRF-Chem model, allowing to process the source properties of ash and $SO_2$ without intermediate steps. Such temporarily and vertically resolved emissions of ash and $SO_2$ can be obtained, e.g., by inverse modelling exploiting

satellite observations. The simply structure of the input data format allows source term characteristics obtained with any suitable method.

Model runs with three emission scenarios were conducted and evaluated for the eruption of the Grimsvötn volcano in 2011. This eruption was unique because ash and $SO_2$ injection heights were separated and a vertical wind shear led to different transport directions of the respective clouds after the eruption. Model performance for ash and $SO_2$ dispersion was therefore

highly sensitive to the source geometry.

The first model scenario neglected different emission geometries of ash and $SO_2$. It used the first observed plume height (15 km) as plume top height for ash and $SO_2$ and assumed constant emission fluxes for the entire eruption period which was estimated to last for 2 days. Emission fluxes were calculated empirically (Mastin et al., 2009) and distributed vertically in a 75/25 umbrella-shaped plume (Stuefer et al., 2013). The second scenario was based on the entire observed plume top height

time series, which was, again, assumed to be the same for ash and $SO_2$. After scaling empirically derived emission fluxes (Moxnes et al. 2014), emitted mass was distributed again in a 75/25 umbrella-shaped plume while considering different plume heights. The third scenario was based on emission fluxes obtained by the inversion of volcanic ash and $SO_2$ column observations from the IASI instrument applying the FLEXPART model to link an a priori source term and satellite total column observations. This source term includes different emission characteristics of ash and $SO_2$, both in the temporal as

well as in the vertical dimension.

Evaluation of the model simulations revealed best performance of the most complex third emission scenario (S3). Improper emission heights of scenario 1 (S1) resulted in overestimated emission fluxes and produced too high ash concentrations.



Furthermore, the ash cloud dispersed into the wrong direction. For the second emission scenario (S2), the simulated magnitudes of the concentrations of ash and $SO_2$ were in good agreement with the satellite observations, although the location of the ash cloud was wrong due to incorrect ash plume top heights, which were in reality lower than those of $SO_2$. This underpins the utility of separate ash and $SO_2$ source terms with reasonable temporal and vertical variability as used in S3. This simulation did not only reproduce the location of ash and $SO_2$ clouds correctly, but also ash concentration values close to the surface.

Validation of simulated vertical ash concentration profiles also revealed a good agreement with observations, although the ash cloud was dispersed already for a few days on the way to the measurement location. The PM10 fraction of the ash could be compared to ground stations in northern Europe. The model underestimates the observations because no other PM10 sources (anthropogenic, biogenic, sea salt …) were considered in the simulations. The prediction of the cloud overpass time was well accomplished for most of the stations by the model run using the complex emission source term S3.

Fast access to on-site measurements, e.g., from volcano observatories is important to constrain dispersion models during an emergency crisis. Decisions must be based on best available information. Updated source term estimates and model hindcasts can help to better understand and predict the transport of ash and gases. Volcanic ash observations from satellite instruments are sometimes limited in accuracy, thus models may help to interpret satellite retrievals. This is crucial for the aviation sector, which is highly vulnerable to "airborne" hazards. Accurate model predictions are not only important to ensure aircraft safety, but can also avoid air space closures or flight re-routings, which can save millions of dollars.

## 6 Acknowledgements

This work has been conducted within the framework of the EUNADICS-AV project, which received funding from the European Union's Horizon 2020 research program for Societal challenges - smart, green and integrated transport under grant agreement no. 723986. This work has also been supported by the BMWFW (Federal Ministry of Science, Research and Economics) through funding of the LUFTLEER project (2020). The publication in part is the result of research sponsored by the Cooperative Institute for Alaska Research with funds from the National Oceanic and Atmospheric Administration under cooperative agreement NA13OAR4320056 with the University of Alaska. Satellite data were made available via the Volcanic Ash Strategic initiative Team (VAST) project web page http://vast.nilu.no/. The authors acknowledge EARLINET for providing aerosol LIDAR profiles available at https://data.earlinet.org/.

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



**Glossary**

| | |
|---|---|
| AATSR | - Advanced Along-Track Scanning Radiometer |
| AGL | - Above Ground Level |
| ASL | - Above Surface Level |
| AIRS | - Atmospheric Infrared Sounder |
| AOT | - Aerosol Optical Thickness |
| DU | - Dobson Units |
| ECMWF | - European Centre for Medium-Range Weather Forecasts |
| EARLINET | - European Aerosol Research Lidar Network |
| ENVISAT | - Environmental Satellite |
| EOS | - Earth Observing System |
| EUNADICS-AV | - European Natural Disaster Coordination and Information System for Aviation |
| FLEXPART | - FLEXible PARTicle dispersion model |
| GOME-2 | - Global Ozone Monitoring Experiment |
| IASI | - Infrared Atmospheric Sounding Interferometer |
| ICAO | - International Civil Aviation Organisation |
| IMO | - Icelandic Meteorological Office |
| Lidar | - LIght Detection And Ranging |
| METEOSAT | - Meteorological satellite |
| MODIS | - Moderate Resolution Imaging Spectroradiometer |
| NOAA | - National Oceanic and Atmospheric Administration |
| OMI | - Ozone Monitoring Instrument |
| PBL | - Planetary Boundary Layer |
| PM | - Particulate Matter |
| RRTMG | - Rapid Radiative Transfer Model for Global radiation schemes |
| SEVIRI | - Spinning Enhanced Visible Infra-Red Imager |
| UTC | - Coordinated Universal Time |
| VAACs | - Volcanic Ash Advisory Centres |
| VACP | - Volcanic Ash Contingency Plan |
| VAST | - Volcanic Ash Strategic initiative Team |
| WRF-Chem | - Weather Research and Forecasting (WRF) model coupled with Chemistry |


**Figures**

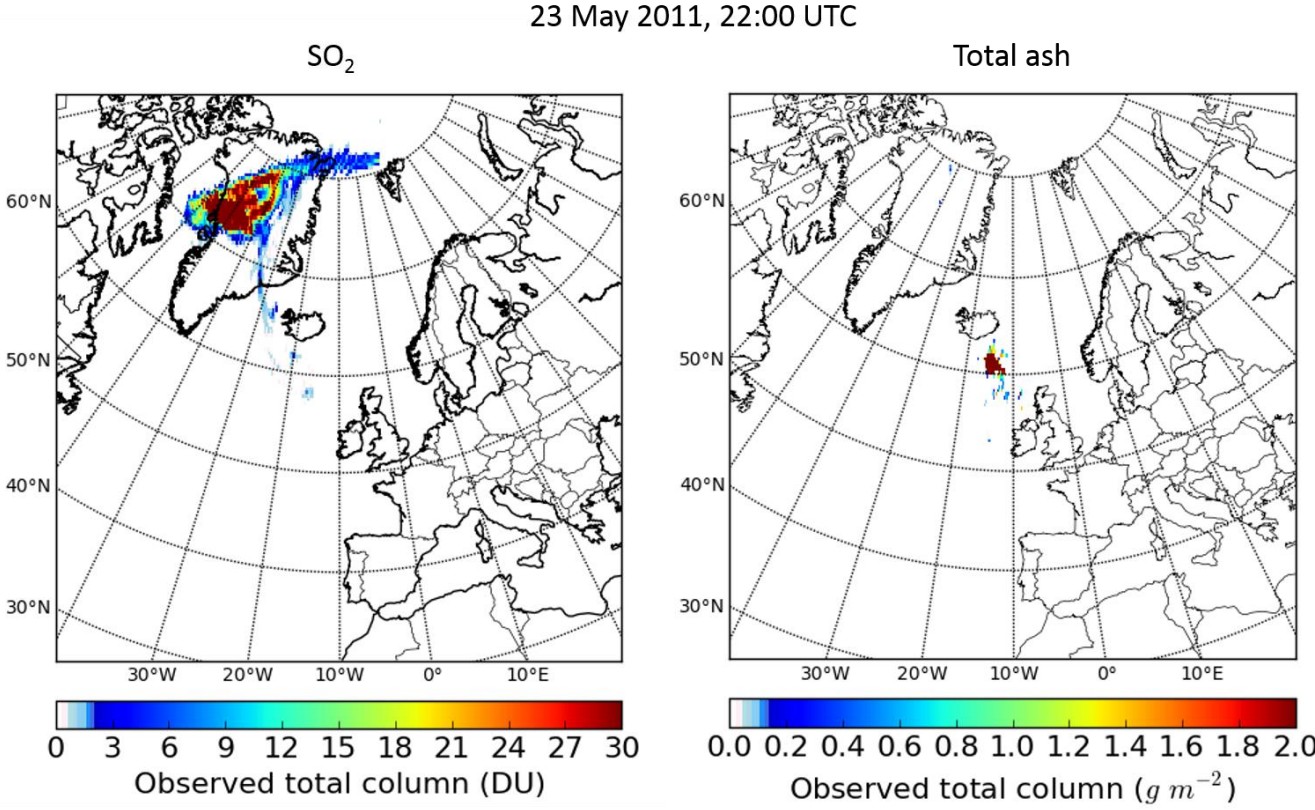

**Figure 1: IASI SO₂ (left) and total ash (right) observations on the 23 May 2011, 22:00 UTC.**





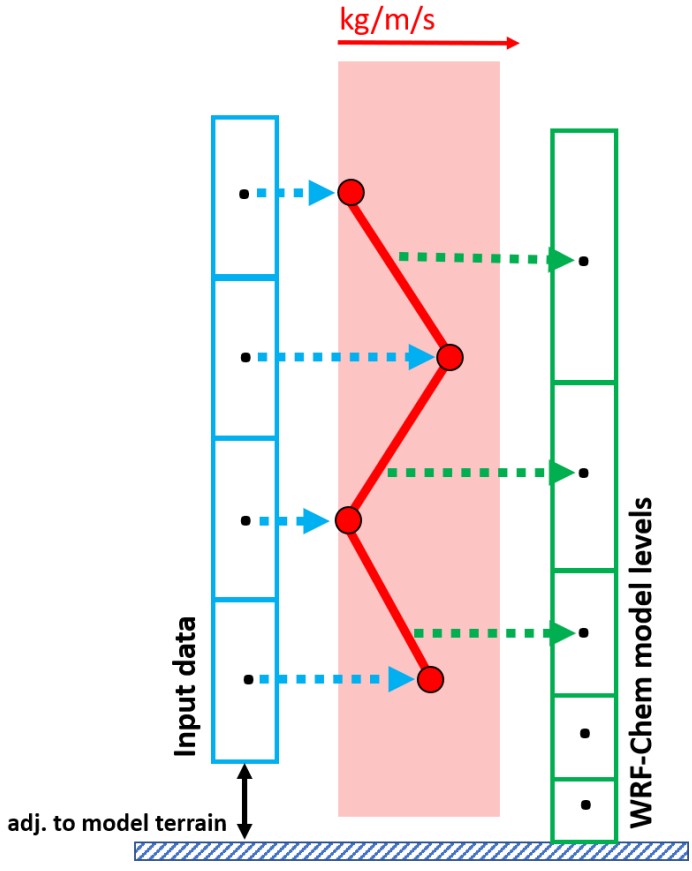

Figure 2: Linear interpolation between input data (blue) and WRF-Chem model levels (green) of the emission flux (red, kg/m/s).

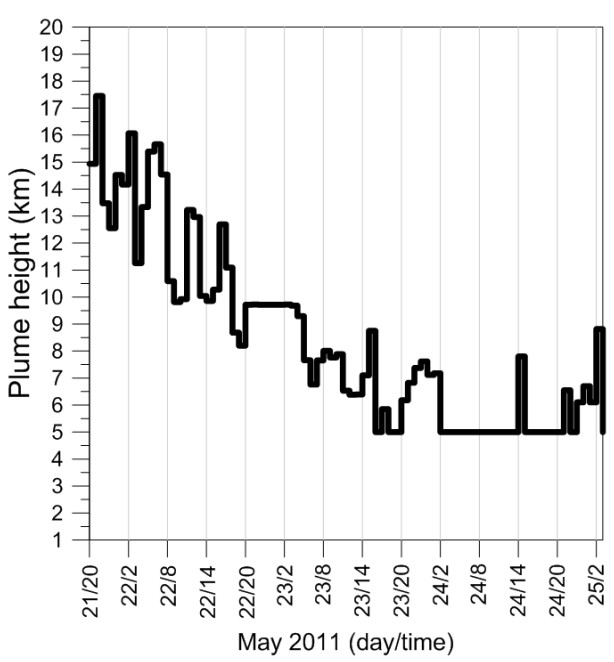


**Figure 3: Observed plume heights (AGL) from the Keflavik radar from the 21 until the 25 May 2011 during the eruption of the Grimsvötn volcano.**

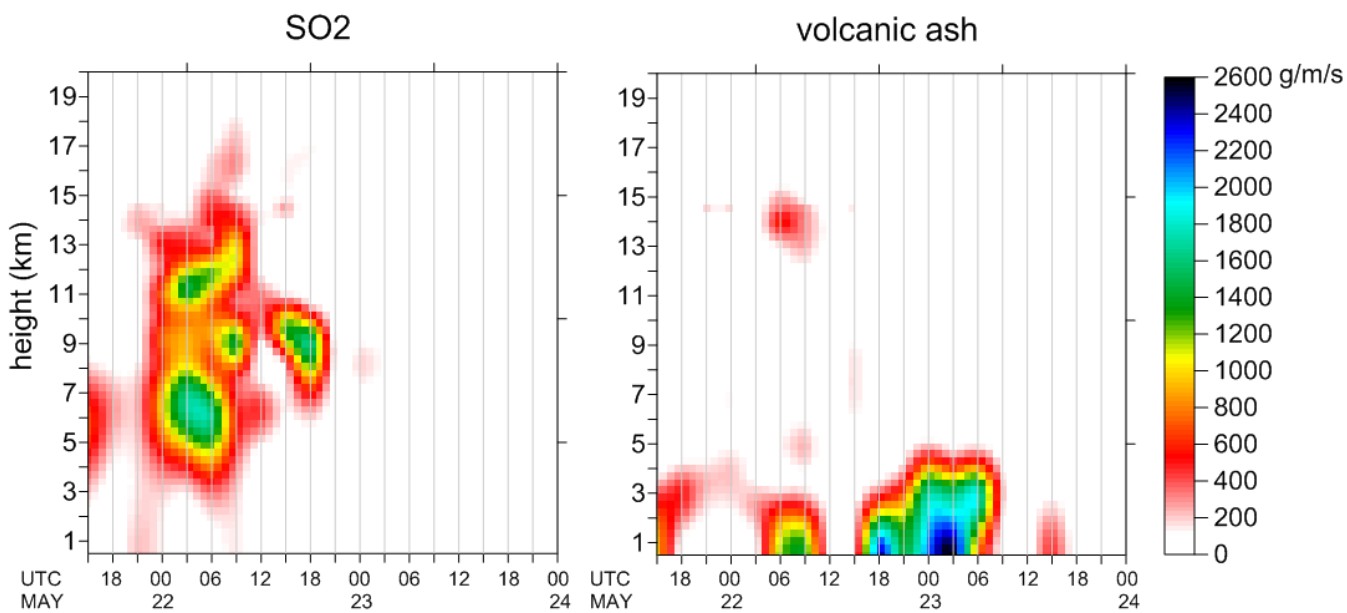

**Figure 4: Temporal evolution of hourly-resolved vertical (height ASL) SO₂ (left) and ash (right) emissions from FLEXPART**
**inverse modelling based on IASI data. Data obtained from Moxnes et al. (2014).**



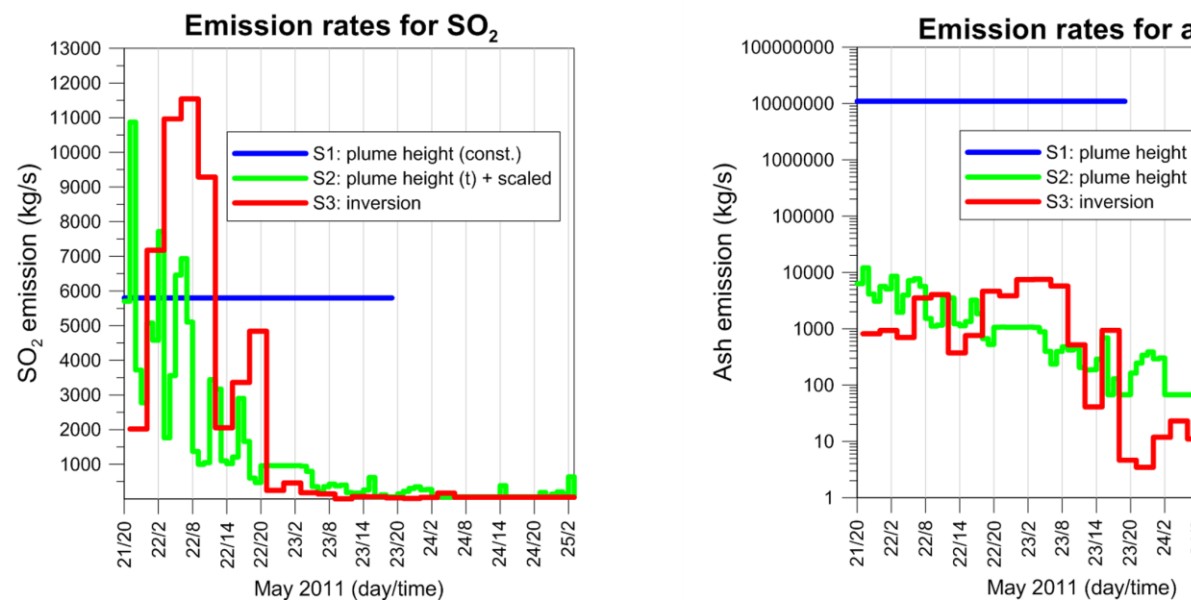

Figure 5: Emission rates for all three emission scenarios for SO₂ (left) and ash (right).
**Figure 6: Maximum sub-column concentrations of total ash indicated via the 4 mg/m³ isoline for each grid cell predicted for the first two days (22 and 23 May 2011, 00:00 and 12:00 UTC) after the eruption start for the three emission scenarios simulated with WRF-Chem.**



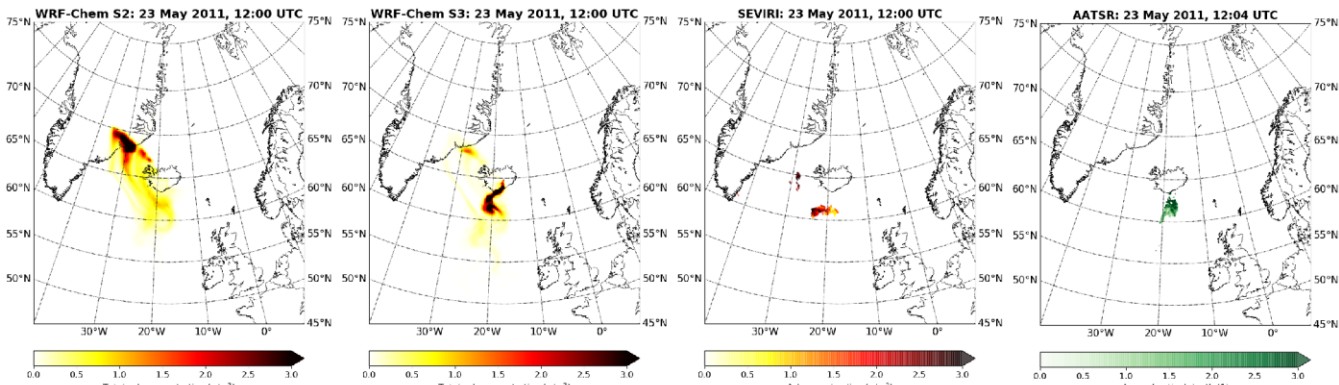

**Figure 7: Total ash columns from WRF-Chem simulations (S2 and S3), SEVIRI ash mass loading and AOT from AATSR.**

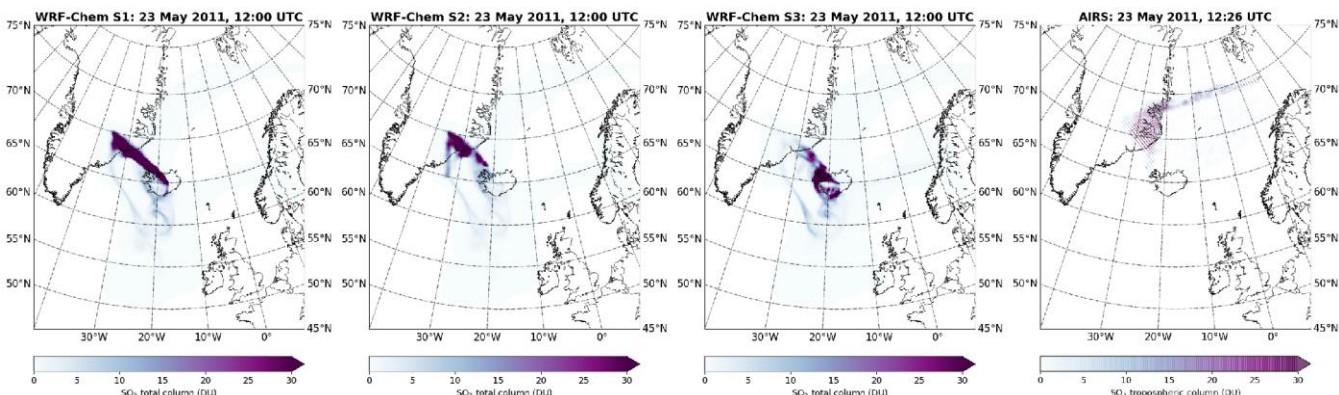

470          **Figure 8: SO₂ total columns (DU) from WRF-Chem simulations (S1, S2 and S3) compared to the AIRS observations.**





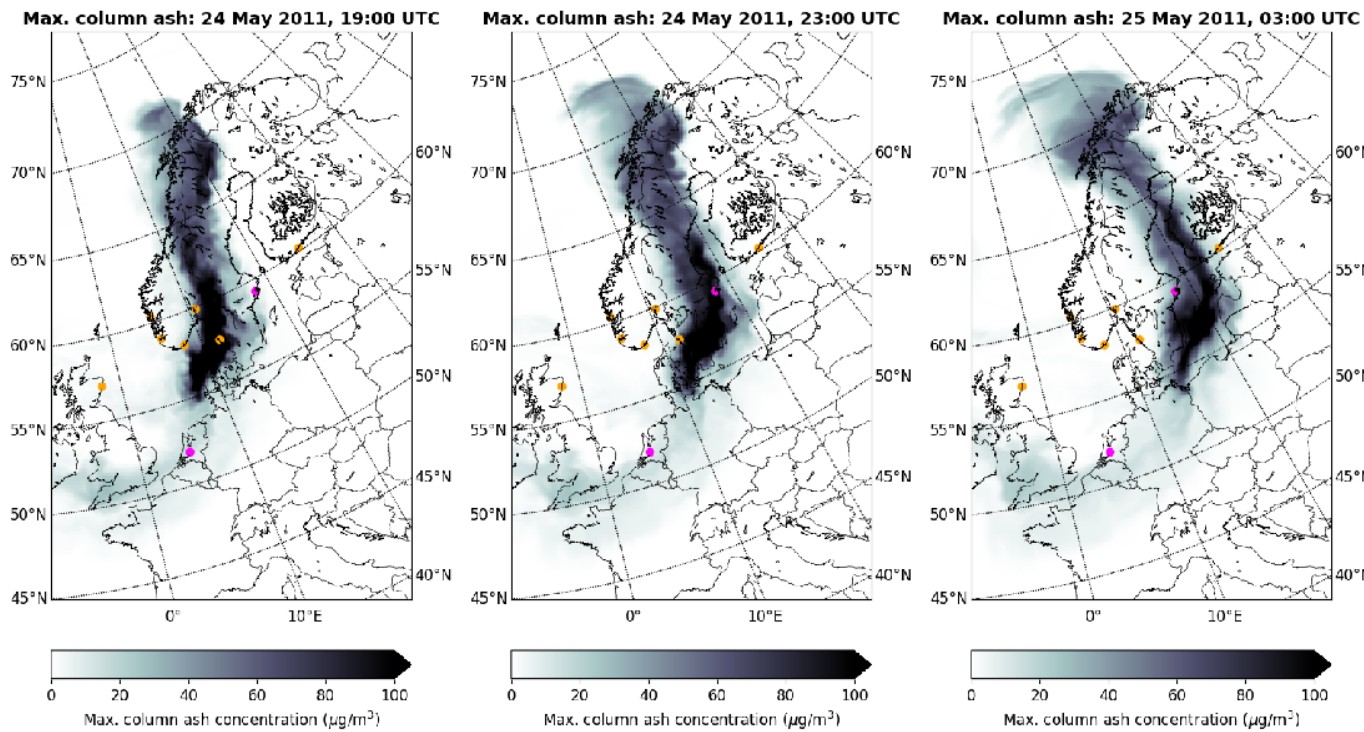

**Figure 9: Simulated maximum total ash column for each grid cell on the 24 May 19:00 (left) 23:00 (middle) and the 25 May 03:00 UTC (right). The pink dots indicate the locations of the lidar in Stockholm and Cabauw, the orange dots indicate the location of the ground stations.**




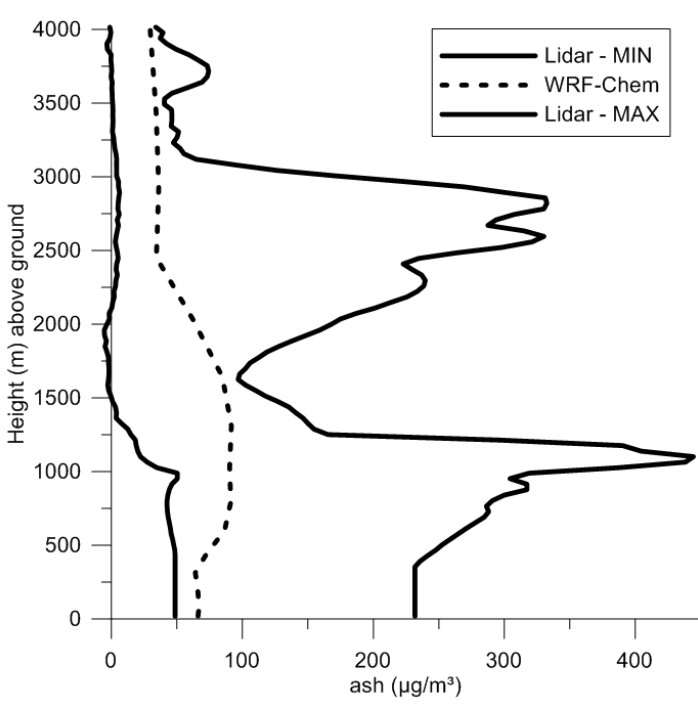


**Figure 10: Min/max observed ash concentrations values at the lidar Stockholm (25 May 02:00 until 08:00) compared to the WRF-Chem maximum ash concentrations (24 May 19:00 until 25 May 03:00 UTC) for each vertical level.**




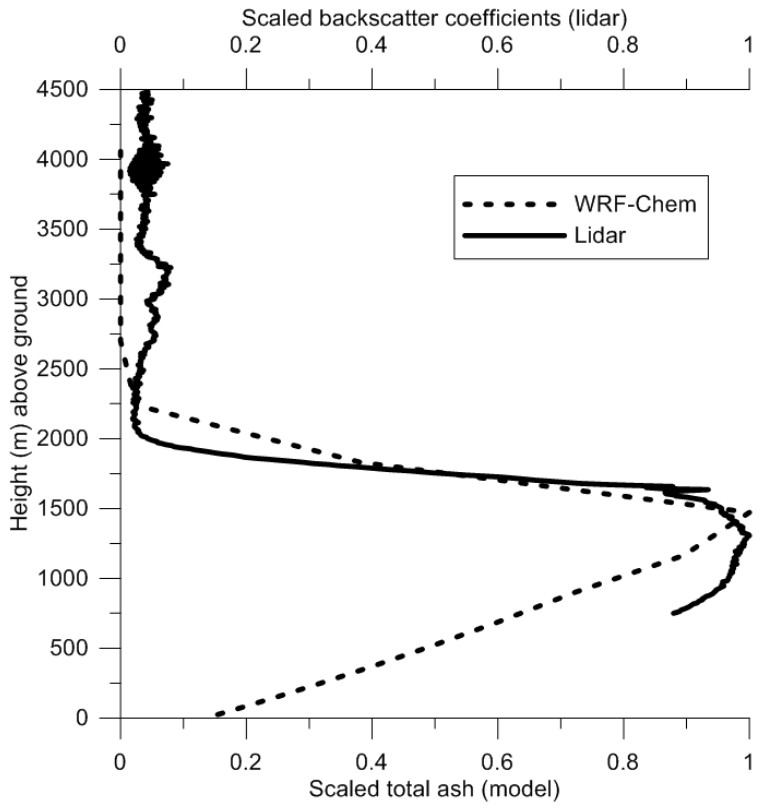

**Figure 11: Vertical (scaled) profiles of WRF-Chem S3-based total ash and backscattering coefficients from the EARLINET lidar at**
**Cabauw on 25 May 2011, 16:30 UTC.**




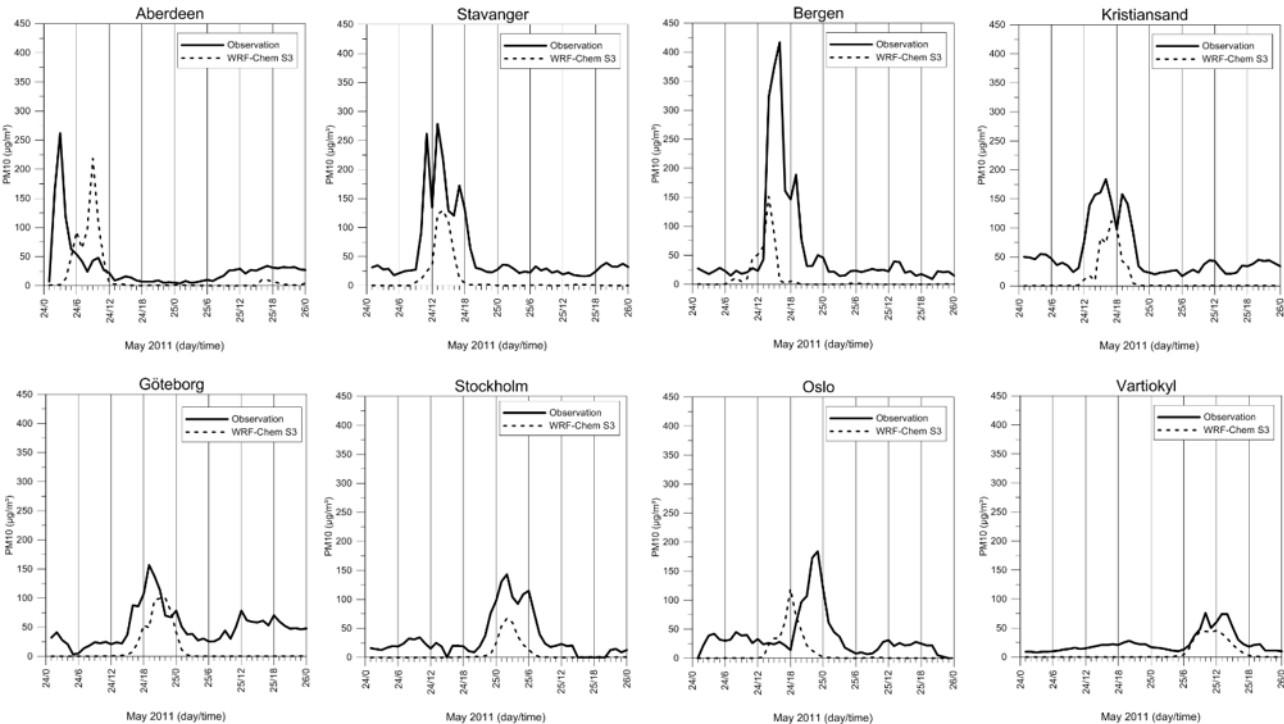

**Figure 12: Time series of observed PM10 (µg/m³) ground concentrations (solid line) and WRF-Chem simulations (dotted line) for 24 and 25 May 2011.**






**Tables**

**Table 1: Example input data from the May 2011 Grimsvötn event for the new volcanic emission pre-processor.**

| Date (in yyyymmdd) | Time (in hhmmss) | Height (AGL in m) | Ash emissions (in kg/m/s) | SO₂ emissions (in kg/m/s) |
|---|---|---|---|---|
| 20110521 | 150000 | 750 | 1.0112 | 0.0002 |
| 20110521 | 150000 | 1250 | 0.9713 | 0.0107 |
| 20110521 | 150000 | 1750 | 0.8887 | 0.0347 |
| 20110521 | 150000 | 2250 | 0.7603 | 0.0748 |
| 20110521 | 150000 | 2750 | 0.0 | 0.0 |
| | | | | |
| 20110521 | 180000 | 750 | 0.0057 | 0.0000 |
| 20110521 | 180000 | 1250 | 0.0753 | 0.0000 |
| 20110521 | 180000 | 1750 | 0.1996 | 0.0000 |
| 20110521 | 180000 | 2250 | 0.3484 | 0.0009 |
| 20110521 | 180000 | 2750 | 0.0 | 0.0 |
| | | | | |
| 20110521 | 210000 | 750 | 0.0000 | 0.2210 |
| 20110521 | 210000 | 1250 | 0.0053 | 0.2135 |
| 20110521 | 210000 | 1750 | 0.0341 | 0.1997 |
| 20110521 | 210000 | 2250 | 0.0897 | 0.1820 |
| 20110521 | 210000 | 2750 | 0.0 | 0.0 |
| | | | | |
| 20110522 | 000000 | 750 | 0.0 | 0.0 |
| 20110522 | 000000 | 1250 | 0.0 | 0.0 |
| 20110522 | 000000 | 1750 | 0.0 | 0.0 |
| 20110522 | 000000 | 2250 | 0.0 | 0.0 |
| 20110522 | 000000 | 2750 | 0.0 | 0.0 |






**Data availability.** Data are available upon request from the corresponding author (marcus.irtl@zamg.ac.at).

**Author contributions.** MH conceptualized and prepared the paper with contributions from all co-authors. MH developed the new WRF-Chem code and conducted the model simulations and data processing for the evaluation with the observational data. MH, BSP and MM collected the observational data from different sources and prepared the figures for the paper. MH interpreted the data with support by BSP and MS. MS and RB supported on WRF-Chem setup and data preparation. CM, DA and MM provided important support for the source term inversion part by providing and interpreting the data which was 510 used for the model scenario S3.

**Competing interests.** The authors declare that they have no conflict of interest.

**Special issue statement.** This article is part of the special issue "Analysis and prediction of natural airborne aviation 515 hazards". It is not associated with a conference.

**Financial support.** This research has been supported by the European Union's Horizon 2020 research programme for societal challenges – Smart, Green and Integrated Transport (grant no. 723986).