# Peer review of "Extension of the WRF-Chem volcanic emission pre-processor to integrate complex source terms and evaluation for different emission scenarios of the Grimsvötn 2011 eruption"

_Natural Hazards and Earth System Sciences, 2020_

## Referee Comment (RC1) · Anonymous Referee #1 · 17 Sep 2020

This is a nice paper on the difficulties to define emission scenarios for volcano eruptions for atmospheric transport modelling explored for the Grimsvötn eruption in 2011. By comparisons with different measurements the merits and deficiencies of three different approaches are clearly demonstrated.

The paper is well written and in good disposition. The paper demonstrates the issues with modelling of volcano eruptions, from forecast into unknown future, feeding in measured plume heights, to hind-cast with assimilated source estimates. The Grimsvötn case also pose the issue of vertical emission split of $SO_2$ and ash.

[Figure]

Figures are appropriate and descriptive. Reference to Carboni et al. 2013 is missing while Carboni et al. 2016 is there, otherwise the references are appropriate.

---

## Referee Comment (RC2) · Anonymous Referee #2 · 25 Sep 2020

General comments:

The manuscript by Hirlt et al., discusses the extension of the volcanic emission module of the chemical transport model WR-Chem. This study presents the update of the WRF-Chem volcanic emission pre-processor towards more complex source terms and evaluates the results for the eruption of the Grimsvötn volcano in Iceland in May 2011. This paper shows that the most complex source term gives better performance. The paper is well-written and I recommend the publication to Nat. Hazards Earth Syst. Sci., after addressing the specific and technical comments listed below.
Specific comments:

In the abstract, "The volcanic emission module of the chemical transport model WRF-Chem has been extended to allow integrating detailed temporally and vertically resolved input data from volcanic eruptions." and then in page 2, line 50-51 "This study presents the extension of the WRF-Chem volcanic emission pre-processor towards more complex source terms and evaluates the results for the eruption of the Grimsvötn volcano in Iceland in May 2011. " Are you updating a module in the WRF-Chem code or a "pre-processor"? I suggest being consistent.

Page 3, line 68-69: In the text, "Forecast models, (. . .) produced unrealistic forecast as shown by comparison to satellite data." Could you explain what do you mean by "unrealistic"? I think you need to justify why you are doing this study. Which are the significant gaps in this field?

I suggest to include a paragraph with the uncertainties in the extension of the volcanic pre-processor (Section 2). For example, what is the uncertainty of "the resulting total volcanic emission, which is used for the WRF-Chem simulation, is scaled; in order to compensate for the deviations caused by the interpolation to ensure mass conservation", and it is scaled to which number? I suggest to rewrite this section in a clear way that give more details about the extension code you have made and its uncertainty.

Section 2, line 85-86: What do you mean by "The online approach (meteorology with air chemistry) accounts for a numerically consistent air quality forecast; no interpolation in time or space is required."? You are not doing "air quality forecast" here.

Page 9, line 264-265: Have you checked if the model is capturing the meteorology for the days of the model simulation?

Conclusions: I am missing a sentence talking about air quality and human health. High concentrations of PM10 and SO2 during a volcano eruption are seen in several regions of Europe (Fig. 12).

[Figure]

Technical corrections:

Page 2, line 37-38: "Low, medium and high" are three regions. However, "less than or equal to 2 mg/m$^3$, greater than 2 mg/m$^3$, less than or equal to 4 mg/m$^3$, and higher than 4 mg/m$^3$" are four groups. Please check that.

Page 6, line 162; "attitudes" Change to: altitudes

Page 6, line 185-186. Please check that all satellites have reference at some point of the manuscript. I don't see any reference for METEOSAT satellite.

Page 7, line 200. Why don't you show 24 May on the Supplementary information if you are discussing the results? Same for the next days from Fig. 8 that is mention in Page 7, line 216.

References: Page 11, line 330 "L. L" and page 13, line 405 "T. V.".

Figures: Figure 3: Title "and" Change to "from".

Figure 12: Axes should be bigger, it is difficult to read. If you are showing only S3, you show specify that at the caption.

[Figure]

---

## Author Comment (AC1) · 1 Oct 2020

Thanks you for the feedback. Concnerning the reference it was a type, I will correct it in the revised version. BR Marcus

---

## Author Comment (AC2) · 9 Oct 2020

*We would like to thank the reviewer for the valuable comments and feedbacks. All the comments are addressed point-by-point below, reviewer's comments are displayed in black, author´s replies in blue fonts.*

**R2:** In the abstract, "The volcanic emission module of the chemical transport model WRF- Chem has been extended to allow integrating detailed temporally and vertically resolved input data from volcanic eruptions." and then in page 2, line 50-51 "This study presents the extension of the WRF-Chem volcanic emission pre-processor towards more complex source terms and evaluates the results for the eruption of the Grimsvötn volcano in Iceland in May 2011." Are you updating a module in the WRF-Chem code or a "pre-processor"? I suggest being consistent.

*The WRF-Chem volcanic pre-processor is integrated into the code of the modeling environment of the WRF-Chem model. In our opinion, the used terms are consistent.*

*We have changed the sentence in the abstract to:*

*The volcanic emission pre-processor of the chemical transport model WRF-Chem …*

**R2:** Page 3, line 68-69: In the text, "Forecast models, (. . .) produced unrealistic forecast as shown by comparison to satellite data." Could you explain what do you mean by "unrealistic"? I think you need to justify why you are doing this study. Which are the significant gaps in this field?

*Comparisons with observational data showed that the assumptions used to estimate the emission source term (constant source parameters) resulted in large differences concerning location and concentration of the ash/$SO_2$ clouds. This has been documented, e.g., by Tesche et al. (2012) and Cooke et al. (2014) which are cited in the text already.*

*In order to justify our paper, we have added the following sentence:*

*The motivation to further develop the volcanic ash emission pre-processor of WRF-Chem was to improve the capabilities of the model to simulate also complex eruption cases.*

**R2:** I suggest to include a paragraph with the uncertainties in the extension of the volcanic pre-processor (Section 2). For example, what is the uncertainty of "the resulting total volcanic emission, which is used for the WRF-Chem simulation, is scaled; in order to compensate for the deviations caused by the interpolation to ensure mass conservation", and it is scaled to which number? I suggest to rewrite this section in a clear way that give more details about the extension code you have made and its uncertainty.

*The uncertainty of the emission that is integrated into the model depends on the source that is used to produce it. No quantitative estimation of uncertainty is possible here. What is done in the pre-processor is that this input data is interpolated to the vertical grid of WRF-Chem.*

*The steps are the following:*

1. *The provided input data is interpolated to the WRF-Chem vertical grid*
2. *The total mass of the input data (MASS_INP) is calculated together with the total mass of the resulting WRF-Chem total column data (MASS_WRFChem).*
3. *Because of the interpolation these two numbers are usually not the same.*
4. *In order to make the WRF-Chem run consistent, the vertical mass fluxes are scaled with MASS_INP/MASS_WRFChem so that the total emitted mass used in the WRF-Chem run is equal to the total mass of the input data*

*We changed the sentence for better readability as follows:*

*Finally, the resulting total volcanic emission, which is used for the WRF-Chem simulation is scaled in order to ensure mass conservation (can be violated due to interpolation effects).*

**R2:** Section 2, line 85-86: What do you mean by "The online approach (meteorology with air chemistry) accounts for a numerically consistent air quality forecast; no interpolation in time or space is required."? You are not doing "air quality forecast" here.

*In on-line coupled models the meteorological- and the air chemistry part are integrated in one numerical modeling system allowing interaction between these two components. That is the advantage compared to the offline approach in which the meteorological- and chemical part are simulated by 2 different models. Offline air chemistry models are driven by meteorological information from an independent model run. A model interface, comprising of reading external meteorological information and interpolating it to the chemistry model grid is required.*

*Since ash and $SO_2$ released by volcanic eruptions can also affect human health, the dispersion of these volcanic clouds is also important for air quality.*

*We revised the sentence, which now reads:*

*The online approach (meteorology with air chemistry) accounts for a numerically consistent air quality forecast.*

**R2:** Page 9, line 264-265: Have you checked if the model is capturing the meteorology for the days of the model simulation?

*The performance of the WRF-Chem model to realistically simulate meteorology during volcanic eruptions has been evaluated for the Eyjafjallajökull 2010 eruption (Hirtl et al., 2019). The focus of this study was the evaluation of different source terms. Since the dispersion of ash is a direct indicator of meteorology (provided a realistic source term), meteorology was indirectly evaluated. A more detailed investigation was beyond the scope of this paper.*

**R2:** Conclusions: I am missing a sentence talking about air quality and human health. High concentrations of PM10 and SO2 during a volcano eruption are seen in several regions of Europe (Fig. 12).

*We have added the following sentence to the conclusions section before the last paragraph:*

*Our analysis showed that volcanic ash can also have an impact on air quality when the cloud touches the ground. Especially for volcanic events which significantly affect surface air pollution, forecast models can support authorities to warn the public.*

**Technical corrections:**

**R2:** Page 2, line 37-38: "Low, medium and high" are three regions. However, "less than or equal to 2 mg/m3, greater than 2 mg/m3, less than or equal to 4 mg/m3, and higher than 4 mg/m3" are four groups. Please check that.

*Thanks for pointing that out. Actually, there are 3 intervals:*

- *<2: low*
- *2<=4: medium*
- *4<=: high*

*We revised the manuscript text, which now reads*

*"less than or equal to 2 mg/m$^3$, greater than 2 mg/m$^3$ and less than or equal to 4 mg/m$^3$, and higher than 4 mg/m$^3$"*

**R2:** Page 6, line 162; "attitudes" Change to: altitudes

*Changed*

**R2:** Page 6, line 185-186. Please check that all satellites have reference at some point of the manuscript. I don't see any reference for METEOSAT satellite.

*We have added:*

*Schmetz, J., Pili, P., Tjemkes, S., Just, D., Kerkmann, J., Rota, S., & Ratier, A.: An introduction to Meteosat second generation (MSG). Bulletin of the American Meteorological Society, 83(7), 977-992, 2002.*

**R2:** Page 7, line 200. Why don't you show 24 May on the Supplementary information if you are discussing the results? Same for the next days from Fig. 8 that is mention in Page 7, line 216.

*For the sake of readability, we only included most relevant figures in the manuscript. However, we will follow this suggestion and upload the figures as supplementary information, e.g.:*

[Figure]

**R2:** References: Page 11, line 330 "L. L" and page 13, line 405 "T. V.".

*The references seem to be OK: "L. L." refers to "L. L. Strow" (L. Larrabee Strow), "T. V." refers to "T. V. Jónsson"*

**R2:** Figures: Figure 3: Title "and" Change to "from".

*Figure 3 title was changed:*

[Figure]

**R2:** Figure 12: Axes should be bigger; it is difficult to read. If you are showing only S3, you show specify that at the caption.

*We have used a larger font for the axis labels. The entry in the legend for WRF-Chem indicates that the model run S3 is depicted. We have also added this information in the caption:*

[Figure]

*Figure 12: Time series of observed PM10 (µg/m³) ground concentrations (solid line) and WRF-Chem (S3) simulations (dotted line) for 24 and 25 May 2011.*